# Major Vessel Segmentation on X-ray Coronary Angiography using Deep Networks with a Novel Penalty Loss Function

**Su Yang**[1]                                                             JYS1153@NAVER.COM
**Jihoon Kweon**[*1]                                                      KJIHOON2@NAVER.COM
**Young-Hak Kim**[*1]                                                  MDYHKIM@AMC.SEOUL.KR
[1] *Department of Cardiology, Asan Medical Center, University of Ulsan College of Medicine, Seoul, South Korea*

**Editors:** Under Review for MIDL 2019

## Abstract

In this study, we proposed a segmentation method of major vessels on X-ray coronary angiography using fully convolutional networks based on U-Net architecture. A novel loss function $pGD$ was introduced by adding a term for penalizing false negative and false positive to generalized dice coefficient ($GD$). DenseNet121 with $pGD$ achieved the highest average DSC of $91.9 \pm 8.7\%$, precision of $91.3 \pm 8.8\%$, and recall of $92.6 \pm 9.6\%$, respectively and showed improved segmentation performance compared to $GD$.

**Keywords:** Deep learning, X-ray coronary angiography, Major vessel segmentation, Penalty Loss Function.

## 1. Introduction

X-ray coronary angiography (CAG) is a primary diagnostic imaging modality for coronary artery diseases. Quantitative coronary angiography (QCA) provides principle morphological indices such as diameter stenosis and lesion length to evaluate coronary lesions. However, QCA analysis shows high inter-observer variability and limited reproducibility with manual correction (Hermiller et al., 1992) because the vessel segmentation of CAG is hindered by several causes (Figure 1). For automated CAG segmentation, although conventional image processing methods (Wan et al., 2018) and deep learning networks (Au et al., 2018; Jo et al., 2019) have been proposed, the segmentation accuracy was not sufficiently high for clinical applications. In clinical practice, QCA analysis still takes 5-10 minutes per image set of a patient for an expert using commercial software with semi-automatic segmentation tool based on edge-detection method.

In this study, a fast and robust method for segmentation of three major vessels is proposed using deep networks with introducing a novel loss function based on generalized dice coefficient (Sudre et al., 2017) to put an additional weight to both false negative and false positive. Five fully convolutional networks were applied to evaluate the segmentation performance by combining simple U-Net (Ronneberger et al., 2015), VGG16 (Simonyan and Zisserman, 2014), ResNet (He et al., 2016), DenseNet (Huang et al., 2017), and InceptionResNetv2 (Szegedy et al., 2017) encoders with U-Net decoder.

---

* Contributed equally

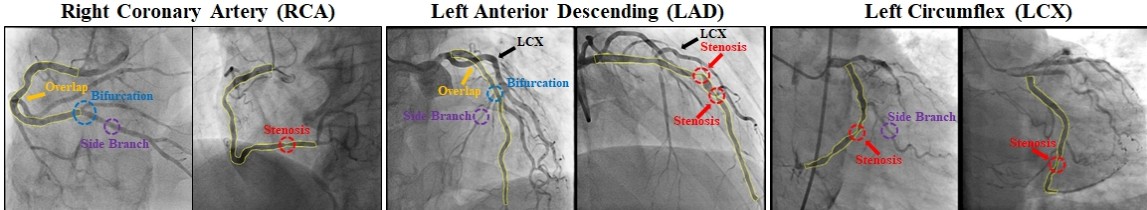

Figure 1: Examples of CAG with segmentation label of major vessels (yellow line). Colored circles show major causes to hinder the automated segmentation of CAG.

## 2. Methods

**Data description.** Patients underwent CAG in Asan Medical Center from February 2016 to November 2016 were retrospectively enrolled. Research approval was granted from Institutional Review Board with a waiver of patient consent. Angiographic images of major vessels for 1980 patients were collected and after excluding the images unable to recognize the coronary structures like total chronic occlusion, a dataset of 5572 images was built. For image labeling, lumen area of major vessel was annotated by two experts using commercial software CAAS workstation 7.5 (Pie Medical Imaging, Netherlands). The ratio of training, validation and test sets was $3 : 1 : 1$.

**Network architecture.** The raw $512{\times}512$ size of CAG images were resized and stacked to 3 channels ($224{\times}224{\times}3$), and the input images were normalized according to 2-dimensional min-max normalization technique. In this work, the network architecture motivated from the U-Net was consisted of deep convolution layers of backbone and five $2 \times 2$ up-sample layers concatenated with skip connection at same level (Appendix A). We used five backbones with initial weights of ImageNet (Russakovsky et al., 2015) for transfer learning.

**Penalty Loss Function.** For binary segmentation, a novel penalty loss function inspired by generalized dice coefficient ($GD$) was introduced. $GD$ is defined as $2(\sum_{l=1}^{c} w_l \sum_{n}^{p} G_{ln} P_{ln})$ $/(\sum_{l=1}^{c} w_l \sum_{n}^{p} (G_{ln} + P_{ln}))$, where $c$ is the number of classes, $p$ is a total number of pixels, $G_{ln}$ is ground truth and $P_{ln}$ is prediction result. $w_l = 1/(\sum_{n}^{p} (G_{ln}/c)^2 + \epsilon)$ is set as a weight to provide invariance to different label properties, where $\epsilon = 10^{-6}$. By adding a penalty proportional to loss, $1 - GD$, $pGD$ is defined as

$$pGD = \frac{GD}{1 + k(1 - GD)} \qquad (1)$$

where $k$ is a penalty coefficient. When $k = 0$, $pGD$ is equivalent to $GD$, while $pGD$ gives additional weights to false positive and false negative for $k > 0$ (Appendix B).

**Training setup.** Mini-batch size was 16 and models were trained for 200 epochs. The data augmentation was performed with rotation ($-20° \sim 20°$), width and height shift ($0 \sim 0.1$), and zoom ($0 \sim 0.1$). For training, Adam optimizer (Kingma and Ba, 2014) was used, and the learning rate which was initially set as $10^{-3}$ was reduced by half up to $10^{-6}$ when the validation loss stayed saturated for 5 epochs on plateau.

## 3. Results

Segmentation performance of deep learning networks was evaluated with dice similarity coefficient (DSC), precision, and recall. In our experiments, DenseNet121 achieved the highest

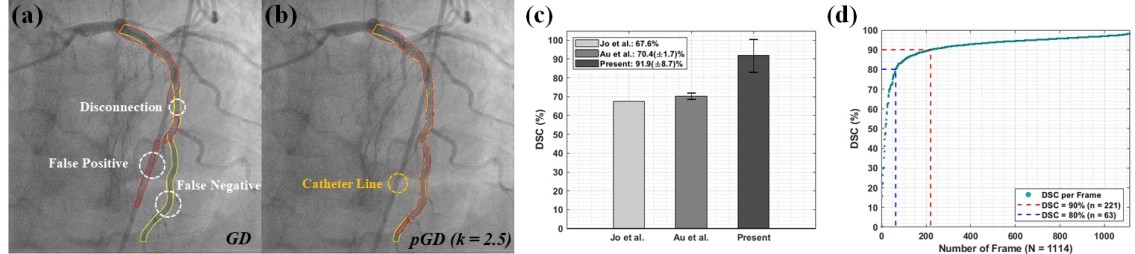

Figure 2: Comparison of ground truth (yellow line) and predicted lumen area (red line) with (a) $GD$ and (b) $pGD$, respectively. (c) DSC comparison with previous reports. (d) Distribution of DSC in test set.

average DSC of $91.9 \pm 8.7\%$ (Table 1). With all the tested networks, $pGD$ showed a higher DSC than $GD$ and the $k$ maximizing DSC was within $2.2 - 2.6$. For DenseNet121, $pGD$ considerably improved the false prediction in the cases of severe vessel overlap or catheter interference (Figure 2). The deep learning required about 0.04 seconds per image.

Table 1: Performance of Proposed Networks (%).    Table 2: Parameter Search of $k$ (%).

| Networks | k | DSC | Precision | Recall | k | DSC | Precision | Recall |
|---|---|---|---|---|---|---|---|---|
| Simple U-Net | 2.3 | $89.8 \pm 9.4$ | $89.8 \pm 9.1$ | $90.3 \pm 10.8$ | 0 | $91.1 \pm 9.7$ | $90.5 \pm 9.9$ | $92.0 \pm 10.7$ |
| VGG16 | 2.2 | $89.3 \pm 10.5$ | $89.4 \pm 10.0$ | $90.0 \pm 12.5$ | 1.0 | $90.7 \pm 10.3$ | $90.1 \pm 10.5$ | $91.8 \pm 11.2$ |
| ResNet101 | 2.5 | $90.3 \pm 9.8$ | $89.5 \pm 10.1$ | $91.5 \pm 10.5$ | 2.0 | $91.6 \pm 8.2$ | $90.9 \pm 8.9$ | $92.6 \pm 9.1$ |
| **DenseNet121** | **2.5** | **$91.9 \pm 8.7$** | **$91.3 \pm 8.8$** | **$92.6 \pm 9.6$** | **2.5** | **$91.9 \pm 8.7$** | **$91.3 \pm 8.8$** | **$92.6 \pm 9.6$** |
| InceptionResNetv2 | 2.6 | $91.7 \pm 9.3$ | $91.0 \pm 9.6$ | $92.7 \pm 10.1$ | 3.0 | $91.7 \pm 8.6$ | $91.4 \pm 8.5$ | $92.3 \pm 9.8$ |

## 4. Discussion

In lumen segmentation of CAG, the present study showed a higher DSC than the previous reports (Au et al., 2018; Jo et al., 2019). The improved predictability was mainly responsible for application of the latest deep learning networks with the large data set augmented with normal vessel images. We further enhanced the segmentation performance with introducing the novel loss function $pGD$. In the lumen prediction with deep learning, over 80% of images in the test set had DSC > 90% and most predictive errors appeared at the proximal or distal part of major vessels, which is little relevant to calculate the quantitative measures of coronary geometry (Figure 3). Deep learning may not only provide a feasible way to be relieved from the QCA analysis that requires labor-intensive manual corrections but also allow an automated real-time diagnostics beyond eye estimation in the clinical practice.

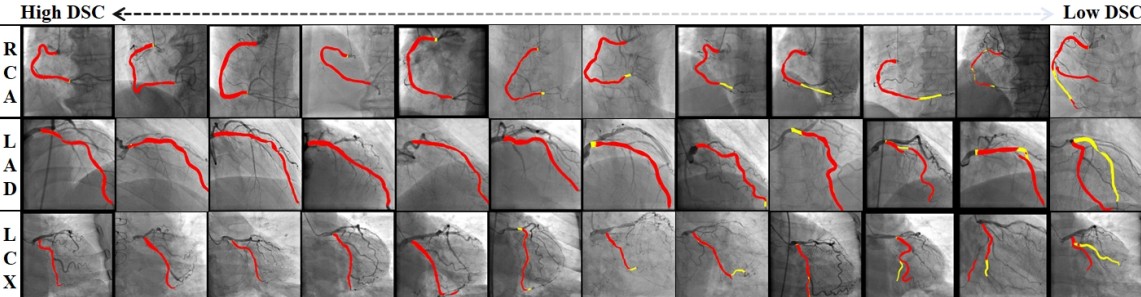

Figure 3: Representative examples of Densenet121 prediction (red) versus ground truth (yellow).

## Acknowledgments

This research was supported by Basic Science Research Program through the National Research Foundation of Korea (NRF) funded by the Ministry of Education (2016R1D1A1A 02937565) and the National Research Foundation of Korea (NRF) grant funded by the Korea government (MSTI) (NRF-2017R1A2B3009800).

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

## Appendix A. Proposed U-Net Architecture with Backbone Encoder.

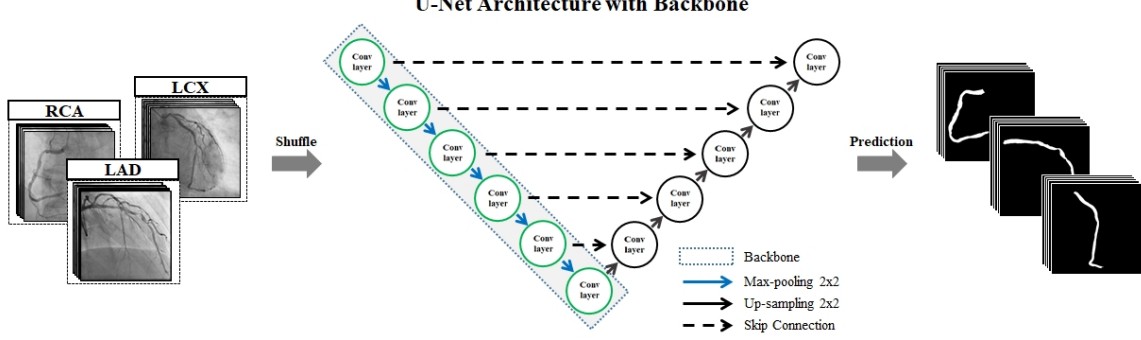

## Appendix B. Proof of Theorem 1

The $\sum_n^p (1 - G_{ln}) P_{ln}$ term is summation of false negative and the $\sum_n^p G_{ln}(1 - P_{ln})$ term is summation of false positive.

$$
\begin{aligned}
pGD &= 2 \frac{\sum_{l=1}^c w_l \sum_n^p G_{ln} P_{ln}}{\sum_{l=1}^c w_l \sum_n^p (G_{ln} + P_{ln}) + k \sum_{l=1}^c w_l \sum_n^p (1 - G_{ln}) P_{ln} + k \sum_{l=1}^c w_l \sum_n^p G_{ln}(1 - P_{ln})} \\
&= 2 \frac{\sum_{l=1}^c w_l \sum_n^p G_{ln} P_{ln}}{\sum_{l=1}^c w_l \sum_n^p (G_{ln} + P_{ln}) + k \sum_{l=1}^c w_l \sum_n^p ((1 - G_{ln}) P_{ln} + G_{ln}(1 - P_{ln}))} \\
&= 2 \frac{\sum_{l=1}^c w_l \sum_n^p G_{ln} P_{ln}}{\sum_{l=1}^c w_l \sum_n^p (G_{ln} + P_{ln}) + k \sum_{l=1}^c w_l \sum_n^p (P_{ln} - 2 P_{ln} G_{ln} + G_{ln})} \\
&= \frac{2 \frac{\sum_{l=1}^c w_l \sum_n^p G_{ln} P_{ln}}{\sum_{l=1}^c w_l \sum_n^p (G_{ln} + P_{ln})}}{\frac{\sum_{l=1}^c w_l \sum_n^p (G_{ln} + P_{ln})}{\sum_{l=1}^c w_l \sum_n^p (G_{ln} + P_{ln})} + \frac{k \sum_{l=1}^c w_l \sum_n^p (P_{ln} - 2 P_{ln} G_{ln} + G_{ln})}{\sum_{l=1}^c w_l \sum_n^p (G_{ln} + P_{ln})}} \\
&= \frac{GD}{1 + k(1 - GD)}
\end{aligned}
\tag{2}
$$

