# OpenReview forum: "Major Vessel Segmentation on X-ray Coronary Angiography using Deep Networks with a Novel Penalty Loss Function"
_MIDL.io/2019/Conference/Abstract — MIDL Abstract 2019_

### Official Review · AnonReviewer2 · 2019-04-29
**A simple and useful extension of the generalised DICE loss for segmentation, with nice empirical results**

**Rating:** 3
**Confidence:** 2

**Review:**

The authors propose pDICE, a generalisation of the generalised DICE score and demonstrated such loss consistently enhances the segmentation performance of various CNN architectures. Also, a form of densely connected networks trained to minimize pDICE was able to achieve very high vessel segmentation performance in X-ray coronary angiography images, an application where the variability in segmentation quality has been a hindrance to subsequent quantitative analysis.

I'm on the boundary between 3. accept and 4. strong accept. Clarification on the questions below would be appreciated.

Questions:
- The generalised DICE [1] extends the standard DICE score by allowing one to assign different weightings to different classes. The proposed pDICE further extends gDICE by allowing one to assign different weightings to FP, FN, TP or TN. Is this understanding correct?

- I also wonder how pDICE relates to the general F score, F_{\beta} (a generalisation of F1 score). We know that minimizing DICE seeks to maximize F1 score. My guess is that minimizing pDICE seeks to maximize F_{\beta} for some value \beta that depends on the value of hyper-parameter k.

[1] Carole H Sudre, Wenqi Li, Tom Vercauteren, Sebastien Ourselin, and M Jorge Cardoso. Generalised dice overlap as a deep learning loss function for highly unbalanced segmen- tations. In Deep learning in medical image analysis and multimodal learning for clinical decision support, pages 240–248. Springer, 2017.

---

### Official Review · AnonReviewer1 · 2019-05-01
**Slightly modified loss function for vessel segmentations from X-ray angiography**

**Rating:** 3
**Confidence:** 2

**Review:**

The authors propose a UNet application to segment X-ray angiography images with a slightly adapted generalized dice loss function.

Pro
- Comparison of multiple approaches with the new loss function
- Results seem to perform better particularly with respect to false positives
- Overall decent results

Cons
- little novelty, very little change to the general dice loss function
- Unclear whether the improved results are generalizeable

---

### Decision · Program_Chairs · 2019-05-06
**Acceptance Decision**

Accept